# Experimental Assessment of the Interface Electronic System for PVDF-Based Piezoelectric Tactile Sensors

**DOI:** 10.3390/s19204437

**Published:** 2019-10-14

**Authors:** Moustafa Saleh, Yahya Abbass, Ali Ibrahim, Maurizio Valle

**Affiliations:** 1Department of Electrical, Electronic and Telecommunication Engineering and Naval architecture (DITEN)-University of Genoa, via Opera Pia 11, 16145 Genoa, Italy; yahya.abbass@edu.unige.it (Y.A.); ali.ibrahim@edu.unige.it (A.I.); Maurizio.valle@unige.it (M.V.); 2MECRL Lab, PhD School for Sciences and Technology (EDST)-Lebanese University, AL Hadath 1003, Lebanon; 3Department of Electrical and Electronics Engineering, Lebanese International University (LIU), Beirut 1105, Lebanon

**Keywords:** interface electronics, tactile sensors, sensors measurements, experimental characterization, signal to noise ratio

## Abstract

Tactile sensors are widely employed to enable the sense of touch for applications such as robotics and prosthetics. In addition to the selection of an appropriate sensing material, the performance of the tactile sensing system is conditioned by its interface electronic system. On the other hand, due to the need to embed the tactile sensing system into a prosthetic device, strict requirements such as small size and low power consumption are imposed on the system design. This paper presents the experimental assessment and characterization of an interface electronic system for piezoelectric tactile sensors for prosthetic applications. The interface electronic is proposed as part of a wearable system intended to be integrated into an upper limb prosthetic device. The system is based on a low power arm-microcontroller and a DDC232 device. Electrical and electromechanical setups have been implemented to assess the response of the interface electronic with PVDF-based piezoelectric sensors. The results of electrical and electromechanical tests validate the correct functionality of the proposed system.

## 1. Introduction

Recent advances in electronic systems are playing a key role in enabling tactile sensing systems to be used for important and critical applications, such as restoring the sense of touch in prosthetics [1]. These applications interact with the surrounding environment through tactile sensors using different transducing methods, such as capacitive [2], piezo-resistive [3], and piezoelectric [4]. Piezoelectric tactile sensors are able to detect contacts and slip, and provide the possibility to estimate applied forces. To meet the application requirements, suitable interface electronic system are needed to amplify and measure the electrical response generated from the tactile sensors. Figure 1 represents the general block diagram of the tactile sensing system. Applied input touch is detected by the sensor, which is then measured and sampled by the interface electronics (IE). Signals are then sent to the processing unit for data decoding e.g., touch modality classifications [5].

When aiming for the reconstruction of accurate contact/touch, such as shape and texture, large numbers of tactile sensors should be deployed [6]. These sensors have to provide high sensitivity, flexibility, and a wide frequency range e.g., polyvinylidene fluoride (PVDF) based piezoelectric sensors [7]. Finally, the frequency of interest for the dynamic interaction range from 0 up to 1 kHz [8]. Unfortunately, few researchers had focused on developing dedicated IE for tactile sensors considering the above mentioned requirements dealing with portable interface electronic systems.

Taking into consideration the described requirements, an interface electronic system integrated with the tactile sensor array has been developed [9]. The system is introduced as a wearable tactile sensing system that can be employed in the prosthetic application. The IE design is based on low power ARM-microcontroller and low-current input analog-to-digital converters that features multiple input channels (i.e., 32 sensors) [9]. In this paper, the experimental characterization of the interface electronics with the piezoelectric PVDF-based tactile sensors is presented. Section 2 reports the state of the art showing the recent developments in IE systems for tactile sensing. In Section 3, the tactile sensing system blocks, along with its implementation, are described. The experimental setup and method for the characterization of the interface electronics with the piezoelectric PVDF-based sensors are illustrated in Section 4. Then, the experimental results are presented in Section 5. After that, the signal to noise and distortion and the effective number of bits of the system are analyzed in Section 6 followed by conclusions in Section 7.

## 2. State of the Art

Interfacing tactile sensors have been given particular importance in research works in the last years. Studies have targeted sensor characterization where the proposed electronics are responsible for collecting sensor signals for analyses and to examine the behavior/response of the sensor [10], [11]. Dedicated interface electronics have been designed for integration purposes in specific applications [9], [12], [13], i.e., robotics and prosthetic hands.

The characterization of PVDF-based piezoelectric sensors has been done by studying their sensitivity with or without protective layers with the variation of the contact force. In particular, when used for acquiring the electrical signals of piezoelectric sensors, the IE was mainly composed of a charge amplifier and a DAQ device for digitizing and filtering the signals. In [10], a prototype of a sensor array, 4 × 2 of a ceramic type, was characterized. The output of the sensors is conditioned using TLV2772 operational amplifier and then sampled through a DAQ device (National instrument, USB-6009). Similarly, the DH5862 device was used in [11] to measure signals of one tactile unit (4 sensors) of a patch with 3 × 2 tactile units (24 sensors). Signals are result of applying three-axis contact forces where normal and shear forces are loaded on the surface of the tactile unit. The output signals, resulting from applying sinusoidal normal force using a shaker in the frequency range 4–500 Hz with variable amplitude in the range of 0–1.5 N, were then collected by an NI-DAQ device (USB-6343). A dedicated IE system was proposed in [11], where the authors presented a design methodology to define the metrics required in assessing the developed IE prototype. The prototype design depends on the sensor charge value that should be detected. So, tests were performed to check the prototype behavior when coupled to integrated PVDF sensors. The IE is composed of an op-amp (OPA348) and a low pass filter. Results reported a sensitivity of about 5.7 pC/N after the PVDF sample had been stimulated with a shaker with fixed frequency (i.e., 230 Hz) at variable force amplitude (i.e., 0.2–0.6 N). Moreover, the IE circuit in [13] adopts a dual-channel analog to digital converter DDC112U and an FPGA Xilinx Spartan^®^-6. The results demonstrated the feasibility of the proposed circuit with 0.6 pC/kPa average sensitivity in the frequency range from 10 Hz to 250 Hz. Furthermore, the functionality of the interface electronics system developed in [13] has been assessed and its implementation was experimentally characterized in [14]. In [9], we proposed a wearable IE system aimed at providing tactile sensory feedback in prosthetic applications. The design provides the possibility to interface 32 piezoelectric tactile sensors with a low power budget maintaining real-time operation.

This paper presents an experimental characterization of our recent developed interface electronics system for PVDF-based piezoelectric tactile sensors. In addition, it describes the system implementation where a low-power microcontroller was adopted to control an analog-to-digital converter of 32 input channels (DDC232 from Texas instrument Inc.). The main contribution of this paper could be summarized as follows:It provides the first experimental assessment of a wearable interface electronics system with PVDF-based piezoelectric tactile sensors.It validates the correct functionality of the design by carrying out electrical and electromechanical tests.It demonstrates the suitability of the proposed system for the prosthetic application when measured charges are analyzed with respect to input forces.

## 3. Tactile Sensing System

### 3.1. The Piezoelectric Tactile Sensor Array

#### 3.1.1. Sensor Structure

Fully screen-printed flexible sensor arrays based on P(VDF-TrFE) (poly(vinylidene fluoride trifluor-oethylene) piezoelectric polymer sensors have been fabricated by JOANNEUM RESEARCH (in the following, JNR) [15]. They patented a low-temperature sol-gel based synthesis for P(VDF-TrFE) inks [16]. Figure 2 shows the structure of a sensing patch built on a sensory array. The fabrication of these sensor arrays is done by screen-printing at a Thieme LAB 1000 [15]. First, a circular bottom electrode is screen-printed on a transparent and flexible (175 μm thick) DIN A4 plastic foil (Melinex^®^ ST 725) substrate. The ferroelectric polymer P(VDF-TrFE) is then screen-printed onto the bottom electrodes, followed by screen printing the top electrodes (Either PEDOT: PSS or carbon have been used as top electrodes [15]). A final UV-curable lacquer layer is deposited on top for overall sensor protection. As a final step, a pooling procedure is then needed to align the thickness direction of the randomly oriented dipoles contained in the P(VDF-TrFE) crystallites.

The very thin thickness of the electrode layer (0.4 µm), with respect to the thicker PVDF-TrFE layer (5 µm), allows for sensor electromechanical modeling, considering its mechanical action is negligible [12], which will be described in the next section.

#### 3.1.2. Sensor Model

The behavior of the piezoelectric sensor is a function of the reaction of piezoelectric transducer layers under an applied contact stress, see Figure 3. Accordingly, the amount of generated charges from the sensor are a function of the amplitude of the applied force. Thus, to estimate such an amount, it is important to have an electromechanical model that shows a relation between the sensor charge and the contact stress. For that, the mathematical mechanical model described in [13] has been adopted. The derived model finds the relationship between the applied mechanical stimulus and the corresponding charge that will be measured by the IE. Equation (1) represents the open-circuit voltage generated by the piezoelectric sensor when a constant vertical stress (T3) is applied:(1)$$V_{oc}\;=-\frac{q_{sensor}}{C_{p}}=\;-\;\frac{d_{33}A_{piezo}}{C_{p}}T_{3}$$
where *q*_sensor_ is the amount of charge generated upon T3 stress, (*d*_33_) is the longitudinal piezoelectric charge coefficient, (*T_3_*) is the mechanical stress, (*C_p_*) is the equivalent capacitance between the electrodes of the piezoelectric film, and (*A_piezo_*) is the loaded piezoelectric area. Since the sensor is covered with a protective layer of thickness (*h*), the direct applied stress is not (*T*_3_).

According to Equation (1), the authors in [13] defined an electrical circuit consisting of a voltage source (*V_oc_*) connected in series with a capacitor (*C_p_*). This represents an equivalent electrical model of the piezoelectric sensor. Thus, the output charge (*q_sensor_*) of the electrical model—equivalent to the output of the real sensors—is calculated in Equation (2) [13]. Furthermore, Equation (2) will be used to calculate the charge generated from the model at the input of the analog-to-digital converter of the IE.
(2)$$q_{sensor\_int}\;=\;C_{p}V_{p}\sin\left(2\pi fT_{INT}\right)$$
where (*f*) is the frequency of the input signal, converted from current-to-voltage every (*T_INT_*) by internal integrators provided by the analog to digital converter (DDC232): Equation (2) will be used as a reference point for the electrical validation of the IE in Section 5.

### 3.2. The Interface Electronic System

#### 3.2.1. Requirements and Specifications

The development of an interface electronic system necessitates possessing quantitative information about the application requirements, such as defining the contact stress/force range and the electrical response of the piezoelectric sensor. These dynamics have been quantified in [17] and can be used as a reference point for defining electronic design specifications. Based on their estimations, the application range goes from 50 Pa to 5 MPa (over 5 orders of magnitude) resulting in a charge response ranging from 0.01 pC to 1 nC. However, the range of interest according to [18] is to cover stresses of the order 10,100 kPa for normal manipulation tasks and those lower than 10 kPa correspond to gentle touches.

Given the considerations above and based on the frequency range of interest for the electronic skin application mentioned in [8], the designed interface electronics should be able to measure an input charge up to hundreds of pC with a large frequency bandwidth up to 1 kHz. Thus, the sampling rate must satisfy the Nyquist condition (above 2 kSps). Moreover, the design must take into consideration a large number of input sensors that will be integrated into the prosthetic glove attached to the amputee’s forearm.

#### 3.2.2. Block Diagram and Circuit Design

This section describes the block diagram and the implementation process for the IE shown in Figure 4, which has been proposed in [9]. The IE comprises two main components: (1) the DDC232 and (2) the BL600. The DDC232 is a 32-channel analog-to-digital converter (DDC232), interfaced at the side of the sensors (32 sensors) with a current offset circuit. The offset circuit is necessary to handle the bipolar charge-output of the sensor that generates both positive and negative charges when stimulated. This can be done by connecting voltage reference equal to (*Vref*/2) at the input of the DDC232 converter. The 32 sensors are connected to the 32 available integrators in the DDC232 so that the current-to-voltage integration can be continuous in time. For the 32 inputs, the output of the 64 integrators are switched to 16 delta-sigma converters via multiplexers where the first 32 integrators are digitized while the other 32 are in the integration mode. The second component is the BL600 module, a low-power ARM-Cortex M0 based microcontroller. It is connected to the ADC using a synchronous serial interface to configure the conversion rate and control the reading process of the converted data [19]. The conversion process is controlled by a CLK pin (configured at frequency 10 MHz) connected to the system clock of the microcontroller. The results of each conversion are stored in a shift register. The output signal (DVALID) goes low to indicate that data are valid and it triggers the controller to start the retrieving process. The retrieved data format can be configured to be either 20 bits or 16 bits. This is done by writing to the 12-bit onboard configuration register the corresponding format value. Three pins DIN_CFG, CLK_CFG, and RESET pins of the ADC are used to write to this register and set the feedback capacitance of the integrators.

The SPI peripheral of the microcontroller has been enabled for controlling the conversion and data retrieval process using the Keil-ARM IDE. The ADC is initialized to convert data with a 16-bit resolution and configured to cover the maximum input charge response by selecting the maximum value of the integrator feedback capacitance, *C_f_* = 87.5 pF; the BL600 module begins the write operation to the register by holding CONV to control the switching between integrators and setting the RESET signal. After that, the configuration process starts by shifting in the configuration data on the configuration register data input (DIN_CFG) on the falling edge of the CLK_CFG (the configuration register clock input). After the configuration is done, a clock signal CLK at a frequency of 10 MHz is generated by the BL600 to operate the ADC. Then, a CONV signal of 1 kHz frequency is generated to convert data of 32 channels at 2 kSps. Sampled data are retrieved by the microcontroller and shifted out on the falling edge of the serial data clock (DCLK) generated at 4 MHz frequency. Finally, data are sent to the PC via UART-to-USB port at a baud rate of 115,200 bits per second (bps) to be collected in the MATLAB tool and further analyzed.

## 4. Experimental Setup and Methods

After design implementation, two experimental tests were performed: electrical and electromechanical. In the electrical test, the equivalent sensor model was implemented by connecting a function generator in series with a capacitor and followed by the interface electronic, as shown in Figure 5. The interface electronics acquire and sample the sinusoidal waveforms produced by the source generator. Then, the sampled data are sent to the PC to reconstruct the original signal where results were verified in Section 5, according to Equation (2).

After the electrical validation of the design, an electromechanical setup was performed to study the functionality of the IE with real sensors. In the electromechanical test, the frequency of the stimulus applied on the sensor has been fixed while the stimulus amplitude was changing by controlling the shaker through a source generator, see Figure 6. The IE connected to the sensor patch measures the generated charges and sends the recorded data to the LabVIEW software. LabVIEW receives the measured applied force from the PCB Piezotronics conditioner as well. In particular, the shaker was controlled to apply 1.2 N as the maximum force with the maximum shaking frequency at 400 Hz. The mechanical chain used for measurements is shown in Figure 7 and it follows the order (up-down). The sensing patch was integrated on a rigid circular substrate and covered by an elastic protected layer (PDMS elastomer layer with thickness 2.5 mm) using double-sided adhesive tape (Model 3M300L, 3M) around the sensor array. The skin patch (sensing patch + PDMS + circular substrate) was then mounted on a fixed support with faced down. The aim of this coupling is building a skin structure that mimics as close as possible real application conditions. A rigid spherical indenter (with radius *R* = 4 mm) and a piezoelectric force transducer (Model 208C01, PCB Piezotronics) were coupled on the moving head of an electromechanical shaker (Brüel&Kjaer, Minishaker Type 4810). All these elements have been accurately aligned before any test. A sinusoidal signal (force) was then provided by a source generator (3390 Arbitrary Waveform Generator) and conveyed to the electromechanical shaker using a Power Amplifier Type 2706. The single taxel was then excited by applying a mechanical stimulus (sinusoidal force) directly on the PDMS patch covering the sensing patch using the shaker. Before running each test, a preload was applied to guarantee indenter-PDMS contact during the whole test. Two tests were done on the same sensors and under the same conditions (same coupling and indenter positioning). In the first test, the charge generated by the sensor was conditioned and processed by the PCB Sensor Signal Conditioner (482C54), while in the second test, the generated charge was processed by the interface electronics. In the two tests, the electromechanical stimulus measured by the piezoelectric force transducer was conditioned and processed by the PCB Sensor Signal Conditioner (482C54). A graphical user interface (GUI) developed with NI LabVIEW on a host PC and NI PCI-4461 DAQ data acquisition board was used to collect and visualize both the force transducer (stimulus) and the generated charge.

## 5. Experimental Results

This section presents the experimental results obtained from both electrical and electromechanical tests.

### 5.1. Electrical Measurement Results

The electrical setup illustrated in Figure 5 was used to validate the interface electronics. An equivalent piezoelectric sensor model, composed of the source generator connected in series with a capacitor of 22 pF, is connected to the IE. The IE was configured to acquire data sampled at 2 kSps. The sampled data was sent to the PC where it was analyzed using MATLAB. The test was done by generating a sine waveform, from the source generator, of a specific frequency at which the IE measures the specified signal at different amplitudes (from 100 mV up to 9 V). The same scenario was repeated at different frequencies within the targeted bandwidth (i.e., 1 Hz–1 kHz). Figure 8a shows that the charge measured with a 100 Hz input sine wave as a function of the amplitude is close to the theoretical results computed from Equation (2). The same test was performed for several frequencies and the results illustrated in Figure 8b verify, according to Equation (2) (*Q_theoratical_ = C_p_V_p_ sin(*2πf*T_INT_)*), that the amount of charge becomes larger as the input frequency increases. Moreover, the non-linearity error was estimated using the best-fit method, where 2.3% for the average error was recorded. Therefore, the results validate the correct functionality of IE in measuring charges, which are close to the theoretical results.

The frequency response of the DDC232 depends on its integrators and the 3-dB bandwidth location is affected according to the integration time. Since the DDC232 is converting data at 2 kSps (means T_integration_ = 500 μs), then according to [19], around 900 Hz bandwidth is within a 3 dB gain. This offers a wide range of frequencies (up to 1 kHz) that can be set for testing. According to datasheet [19] and based on the fact that the DDC is a current-input analog to digital converter, the maximum input current that can be acquired is around 216 uA. Thus, at the maximum current, the signal is at the digital full-scale and any further increase in signal amplitude results in an error. For this reason, the error was realized when the injected signals of high frequencies (above 400 Hz) reached amplitudes above 6 V. Therefore, DDC will reconstruct signals up to 1 kHz frequency at amplitudes up to approximately 3 V. So, high-frequency signals at high amplitudes were not included in the test.

### 5.2. Electromechanical Measurement Results

Figure 9a shows the behavior of the IE in detecting and measuring the change of charges as a function of force. Although the obtained results meet the concept expressed in Equation (2) that as force increases charge increases, the conditioner was included in the test to be a reference point for evaluating the behavior of the IE. In addition to force conditioning, the conditioner features charge conditioning from piezoelectric sensors. Then, the IE was disconnected from the sensors and replaced by a conditioner. The test was repeated under the same conditions (sensor coupling and positioning) and the corresponding force and charge measurements were recorded. It is noticed, as expected from Figure 9b, that when increasing the force, an increase of the amount of charge is observed. We can deduct from Figure 9 that the IE is able to detect and measure the linear change of input charges within the given frequency and force ranges, 20–350 Hz and 0.2–1.2 N, respectively, with slight differences in charge values. The difference in charge values between IE and conditioner can be observed from the sensitivity curve reported in Figure 10a. The plot demonstrates the increase in sensitivity (1.5–31.96 pC/N) as a function of frequency. The sensitivities have been estimated by calculating the slopes of the measured charge versus force within the frequency range 20–400 Hz. By analyzing the figure, it seems that the difference between the IE and the conditioner is almost constant and, thus, it is possible to apply setup calibration to find the empirical difference value. Moreover, in order to demonstrate the ability of the IE in acquiring data at higher frequencies, a test was done by fixing the frequency at 400 Hz and adjusting the amplitude—controlling the shaker—to reach the minimum force value. Results presented in Figure 10b show that the IE is able to measure charges obtained under minimum applied force (0.01 N).

## 6. Signal to Noise Ratio Analysis

This section presents the analysis carried out to study the measured signal with respect to noise by using methods as adopted in [14]. The noise may be a result of harmonic distortion added to the input signal or at the output of the IE circuits described above as we are analyzing the signal output of the whole system, i.e., sensors and interface electronics. Such noise contributes directly to the signal-to-noise ratio of the design. IEEE in [20] has defined test methods for analog to digital converters, i.e., the signal to noise and distortion ratio (SINAD) and the effective number of bits (ENOB). In particular, SINAD is used to measure the degradation of the signal by unwanted signals in noise and distortion. It is the ratio of total signal power level to the noise and distortion power. Also, SINAD provides the basis for calculating the ENOB, which specifies the number of bits of the signal that is above the noise floor. Figure 12 illustrates the SINAD and ENOB curve versus frequency where SINAD was computed by calculating the ratio of the root-mean-square (rms) of the fundamental signal to the root-mean-square of noise and distortion. After normalizing the input data to scale between 0 and 4.09 V (ADC reference voltage), FFT has been applied to distinguish the fundamental signal from other harmonics and noise existing in the spectrum. Finally, the amplitude of the signals in the spectrum have been measured and substituted in the equation below to calculate the SINAD:(3)$$\mathrm{SIN}\mathrm{AD}\;=\;20\;\log\frac{\mathrm{rms}\left(\mathrm{As}\right)\;}{\mathrm{rms}\left(n+k\right)}$$
where As is the amplitude of fundamental signal and (n + k) is the amplitude of noise and distortion. Moreover, we used Equation (4), derived in [21], to compute the ENOB after applying a fast Fourier transformation (FFT) to the data recorded in the previous test. Figure 11 shows an example of an input signal at 100 Hz: (a) original signal in the time domain, (b) reconstructed signal after ADC conversion, and (c) the FFT of the signal demonstrating the fundamental signal at 100 Hz of frequency.

(4)$$\mathrm{ENOB}\;=\;\frac{SINAD-1.76\;dB+20\log\left(\frac{Full\;scale\;amplitude}{Input\;amplitude}\right)\;}{6.02}$$

Figure 12 shows consistent behavior of the IE over a frequency where around 14 bits out of 16 of the digitized signal are above the noise floor. So, this will be an advantage to retain more accurate data and, thus, acquiring tactile data with a high resolution.

## 7. Discussion and Conclusions

This paper presented an experimental assessment of new wearable interface electronics proposed for interfacing PVDF-based piezoelectric tactile sensors for prosthetic application. Interface electronics is composed of a low-power ARM-Cortex M0 microcontroller and a DDC232 analog-to-digital converter to interface 32-input tactile sensors. The system has been experimentally evaluated by electrical and electromechanical tests where results demonstrate charge estimation within the force range 0.01 to 1.2 N (10 kPa–1.4 MPa) with approximately 0.2 pC charge value readable by the IE at 0.01 N, lowest input force. Results validate the proper functionality of the interface electronics in measuring the dynamic range of charges estimated within the force range of interest to cover normal manipulation tasks and stresses of orders up to 100 kPa. An average signal-to-noise and distortion ratio of about 56 dB was measured for applied forces from 0.2 to 1.2 N. Therefore, the results demonstrate the suitability of the proposed system for acquiring tactile bipolar signals with high-resolution after achieving 14 bits of ENOB.

## Figures and Tables

**Figure 1 sensors-19-04437-f001:**
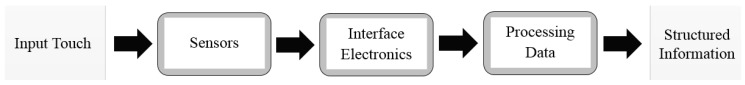
Block diagram for the tactile sensing system.

**Figure 2 sensors-19-04437-f002:**
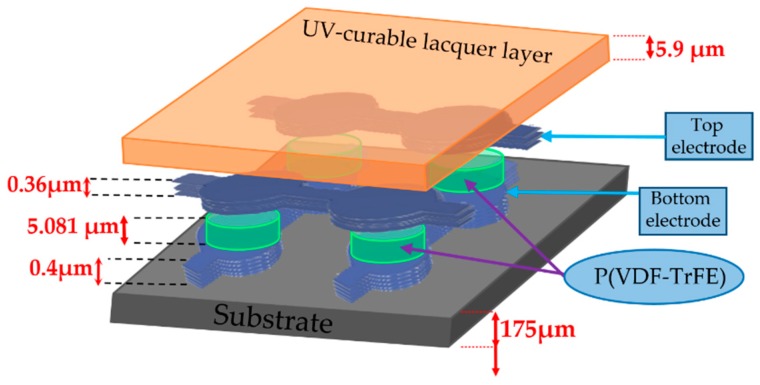
Cross-sectional view of a single sensor unit, sketch with indicative thicknesses of the various layers.

**Figure 3 sensors-19-04437-f003:**
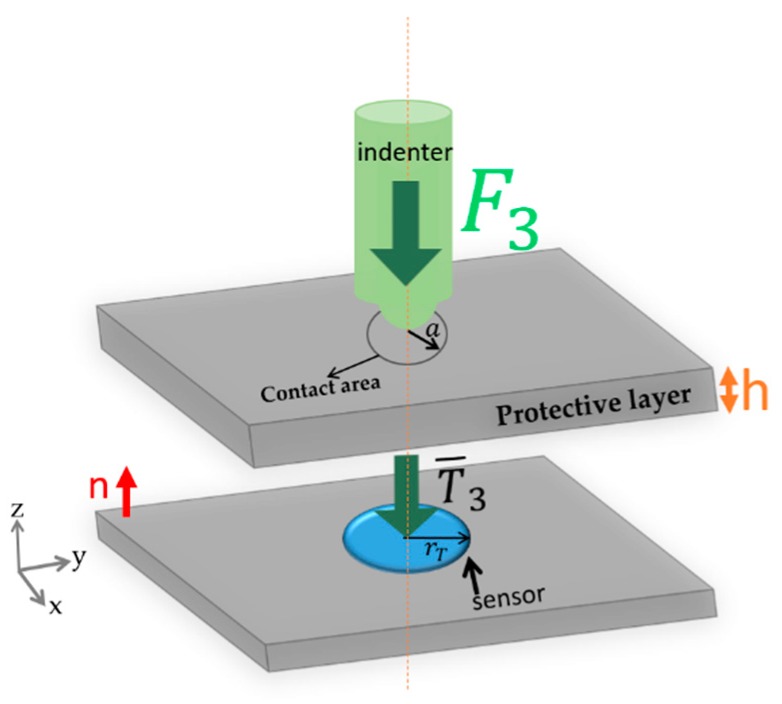
Sketch of the general working mechanism of P(VDF-TrFE) sensors.

**Figure 4 sensors-19-04437-f004:**
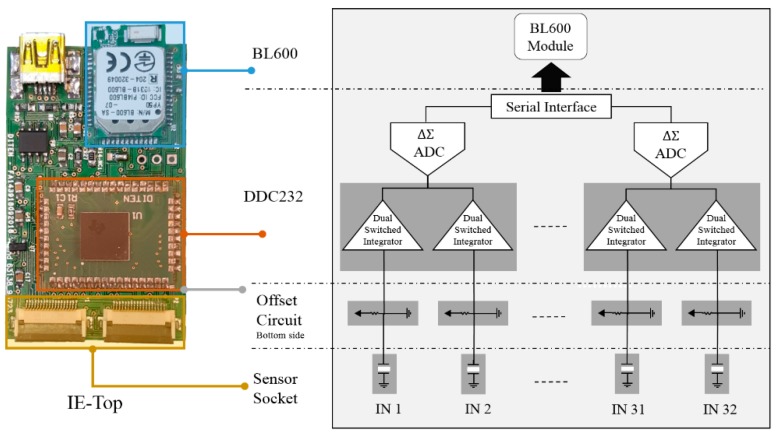
Interface electronic printed board circuit (left) and block diagram of the DDC232 (right) [9].

**Figure 5 sensors-19-04437-f005:**
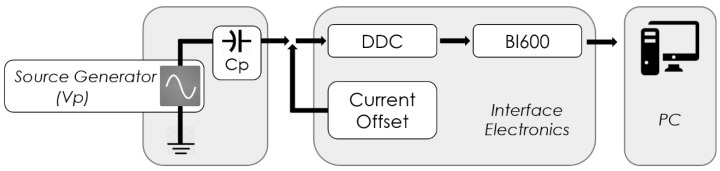
Block diagram of the electrical setup; equivalent circuit of sensor (left) connected to interface electronics; generated signals are reconstructed by the interface electronics (IE) and sent to the PC.

**Figure 6 sensors-19-04437-f006:**
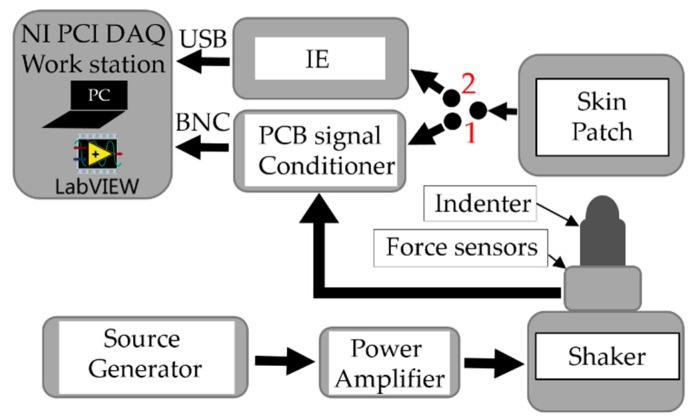
Experimental setup block diagram.

**Figure 7 sensors-19-04437-f007:**
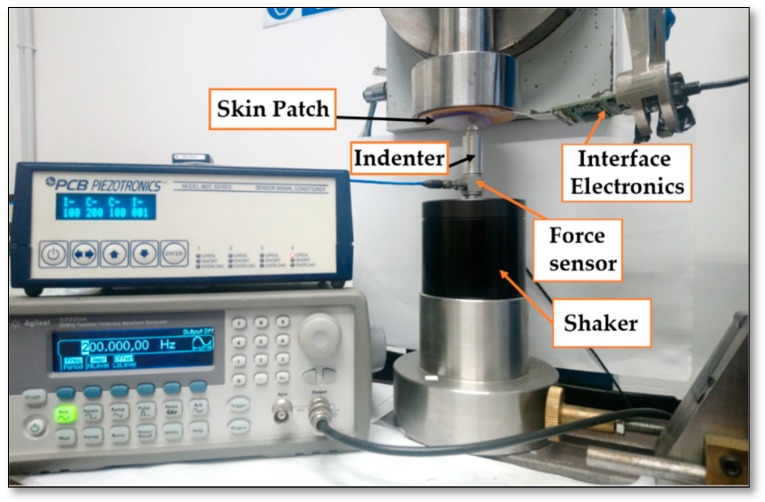
Experimental setup.

**Figure 8 sensors-19-04437-f008:**
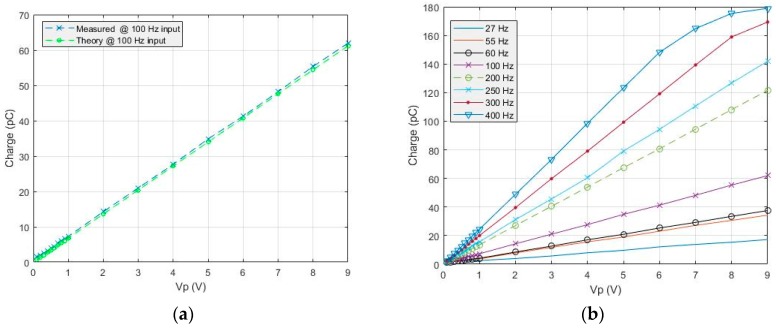
(**a**) The theoretical fit-line is calculated from *Qtheoretical = CpVp sin(*2πf*T_INT_)* derived from the equations presented in [13]; (**b**) shows the output of the IE relative to input signals generated from the source generator.

**Figure 9 sensors-19-04437-f009:**
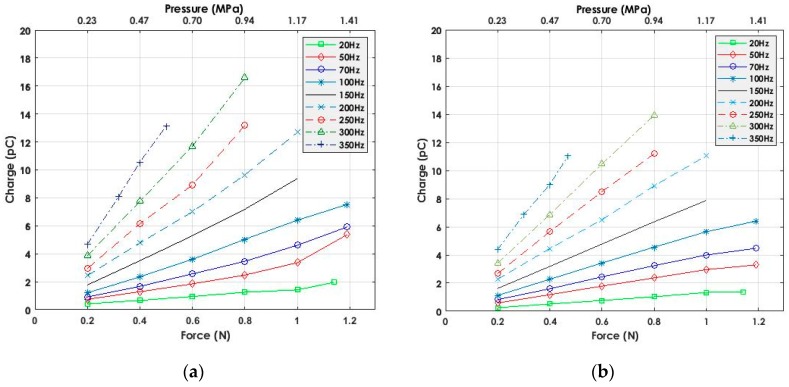
(**a**) IE output measurements with real sensors; (**b**) conditioner output measurements with real sensors.

**Figure 10 sensors-19-04437-f010:**
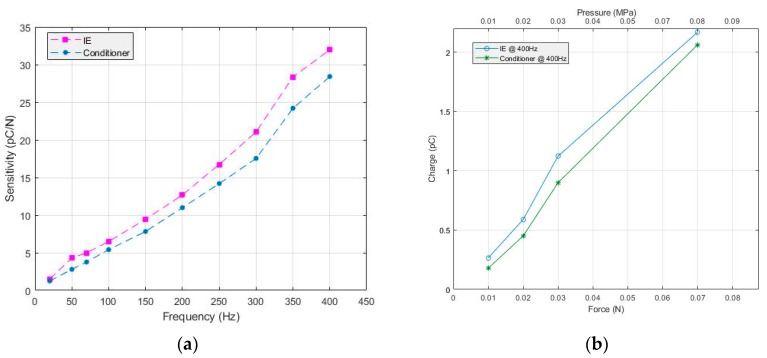
(**a**) Sensitivity as a function of frequency; (**b**) measured charges at the minimum detectable force (0.01 N).

**Figure 11 sensors-19-04437-f011:**
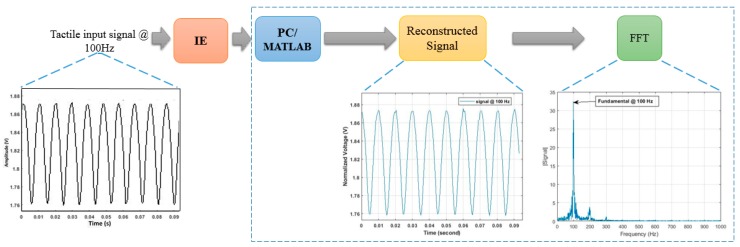
Example of an input signal at 100 Hz of frequency in time and frequency domains.

**Figure 12 sensors-19-04437-f012:**
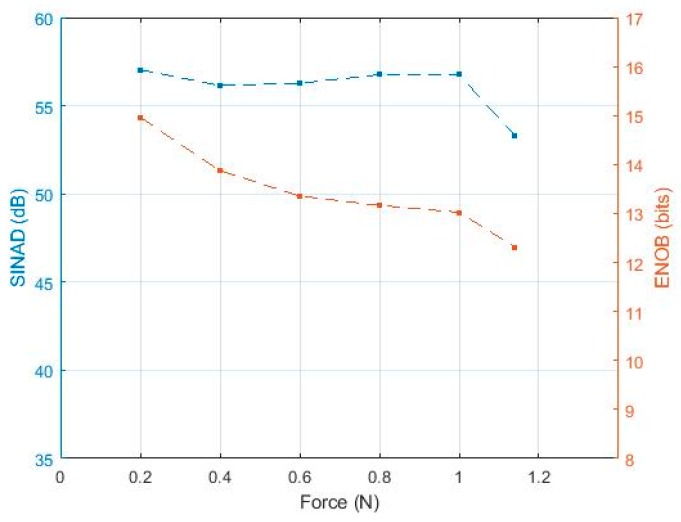
SINAD and ENOB variation with respect to applied forces.

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
