# Peer review of "Experimental Assessment of the Interface Electronic System for PVDF-Based Piezoelectric Tactile Sensors"

_sensors, 2019, doi:10.3390/s19204437_

Round 1

Reviewer 1 Report

The authors present an experimental assessment of an interface electronic (IE) for PVDF-based tactile sensors oriented for robotic and prosthetic applications. The authors submitted a clear and detail experimental description of electrical and electromechanical characterization.

The suitability of the circuit have been well demonstrated, but there are some aspects that are not clear in this research that I consider important to take into account in order to improve the quality of work: 

Comment 1: In the electrical tests, I do not see clearly how the authors estimate the charge (Figure 6) from a function generator connected directly to the IE input. I think it is convenient to add a figure that shows the electrical connections made in these tests.

Comment 2: In line 225, the authors wrote: “However, signals at high amplitude (9 V: the max peak voltage that can be generated by source generator), with frequencies applied above 400 Hz leads to full-range voltage at the DDC232 input. So, these signals were not included in the test”

This means that the bandwidth of the interface electronic is 400 Hz? I consider that it is important to carry out tests that allow to estimate the frequency response of IE in order to experimentally determine the -3 dB frequency.

Comment 3: Why the authors did not present an estimate of the non-linearity errors?

Comment 4: I think that the text from line 230 to 235 (... at 400 Hz.) and Figure 7 should be included in section 4 "Experimental setup and methods".

Comment 5: In section 7: Discussion and conclusion, the authors wrote “…14 bits of ENOB demonstrating its suitability for the prosthetic application.” How can authors assert this if in this work no tests were done on applications of this type?

  In some paragraphs of the document there are some errors that must be corrected.

Line 40: the authors wrote: “[7]. And finally,…” I think it must be: …[7]. Finally,…. Line 110: The authors used P(VDF-TrFE) in all the document. Is PVDF-TrFE correct? Line 128: (qsensor) instead of (Qsensor). Line 129: equation (2) instead of equation 2. Line 273: …equation (5) below… instead of “formula below (5). Line 276: ADC instead of ACD. Line 319: Reference 9 does not have the name of the journal/proceedings.

Author Response

We would like to thank the reviewer who had dedicated time and efforts to review the paper in order to improve its quality. We have applied the requested modifications on the paper responding to the reviewer comments.

Reviewer 2 Report

- Edition should be corrected. Units should be consistently separated by a space (except %). I suggest to use “hard space” to separate them. - I think that all symbols and variables should be explicitly described in the places where they are introduced for the first time in the paper. Thus, parameters Qsensor in (1) also should be described. - Variables should be in italic. Why are variables once enclosed in brackets and at other times not (Chapter 3.1.2)? Please also edit correctly subscripts in this chapter. - Equation (2) is directly taken from [13, eq. (4)]. Thus, the paper [13] should be quoted before this equation. - Equation (2) should also match the text in this paper. What a variable should be used in it: Qsensor or qsensor_int? - The variable w is not the signal frequency. It is an angular frequency of this signal w = 2*pi*f. You wrote it before eq. (1) in [13]. - You use the BL600-SA Bluetooth 4.0 module which includes an ARM Cortex M0 microcontroller. This more detailed information should be included in Chapter 3.2.2. It should also clearly write that IE has been proposed and studied in the paper [9]. - In the caption of Figure 4 there should be a reference to [9] from the same reason. - Lines 178 and 179. Where are data sent? Are they sent to the PC? - Lines 185 and 186. There is nothing to compare to in Chapter 3.1.2. In this chapter there are two equations and no theoretical results. - Exactly which NI DAQ data acquisition board model was used? - Equation “Qtheoratical = CpVp sin(w*TINT)” should be written in italic in the text. What is difference between Qtheoratical, Qsensor or qsensor_int? - Axes “Charge (pC)” in the graphs in Fig. 8 should be scaled to a range of 0-20 pC. - The novelty of the paper is included in Chapter 5. Therefore a relative error graph with conclusions could be used to evaluate the IE. In this case, the PCB signal conditioner can be treated as a reference device. Thanks to this the readers will obtain a fuller description of the proposed solution in this paper. - Chapter 6 is largely taken from chapter Chapter 6 in [14]. This fact should be also mentioned. - Symbols of variables should be uniform in the whole paper. That is, a given symbol should have only one clear meaning in the whole paper. For example: in line 124 “h” is the thickness, but in line 272 “h” is the amplitude of distortion. - In conclusion - the most important note: the authors should more underline what is the novelty of the paper.

Author Response

(The authors gave the same response as above.)

Reviewer 3 Report

The authors studied the interface sensing system using PVDF based piezoelectric sensor. The analysis makes sense and the experimental setup support their argument. The paper is acceptable for publication in Sensors.
I have one question. Although the authors suggest that such sensing system is intended for prosthetic applications, the experimental design has nothing to do with such application. For prosthetic application, the sensing range and response time is only one part of the consideration, the biocompatibility of the sensor material, the bending reliability when attache to the curved surface of the prosthetic, and robustness of the sensor should also be considered. Please address such quotation and modified the abstract and introduction accordingly.

Author Response

(The authors gave the same response as above.)

Round 2

Reviewer 1 Report

The authors addressed all the suggested comments. The paper has improved in this new version. I recommend publishing this work in the present form.

Reviewer 2 Report

The new version is definitely an improvement over the original one.

I don't have any remarks.